# Muscle Fatty Acids, Meat Flavor Compounds and Sensory Characteristics of Xiangxi Yellow Cattle in Comparison to Aberdeen Angus

**DOI:** 10.3390/ani12091161

**Published:** 2022-04-30

**Authors:** Dong Chen, Xiaoyan Wang, Qian Guo, Huifen Deng, Jie Luo, Kangle Yi, Ao Sun, Kun Chen, Qingwu Shen

**Affiliations:** 1College of Animal Sciences, Hunan Agricultural University, Changsha 410128, China; chendong_326@126.com (D.C.); wxyzxs21@126.com (X.W.); 2College of Food Science and Technology, Hunan Agricultural University, Changsha 410128, China; gq3244772628@163.com (Q.G.); luojie@hunau.edu.cn (J.L.); 3Hunan Biological and Electromechanical Polytechnic, Changsha 410127, China; deng_huifen@163.com; 4Hunan Institute of Animal and Veterinary Science, Changsha 410130, China; yikangle@yeah.net; 5Hunan Aite Agriculture and Animal Husbandry Technology Development Co., Ltd., Changsha 410128, China; chenkun1156495776@163.com

**Keywords:** Xiangxi yellow cattle, fatty acids, volatile compounds, meat quality

## Abstract

**Simple Summary:**

With agricultural mechanization and the increasing demand for meat, Xiangxi yellow cattle are increasingly being raised for meat production. Little is known about the changes in muscle composition and eating quality of the cattle at different ages. In the present study, the muscle proximate composition, meat flavor substances, and sensory quality of Xiangxi yellow cattle at different growth stages were profiled in comparison to Aberdeen Angus cattle, which may provide valuable information for high grade beef production from Xiangxi yellow cattle. This study showed that Xiangxi yellow cattle is a fine cattle breed with equal or even better meat quality attributes when compared to Angus. It is proper to slaughter Xiangxi yellow cattle at the age of 18 months for high quality beef production.

**Abstract:**

The objective of this study was to investigate meat quality of Xiangxi yellow cattle of different ages in comparison to Aberdeen Angus. At the ages of 6, 18, and 30 months, 10 female animals for both Xiangxi yellow cattle and Aberdeen Angus cattle were randomly selected and slaughtered. The proximate composition analysis, fatty acid profiles and flavor compounds were measured on the *longissimus thoracis* (LT) muscle samples. One boneless loin chop was dissected and used for sensory evaluation by a 10-persoon trained taste panel. The data obtained showed that Xiangxi yellow cattle deposited similarly high level of intramuscular fat as Angus at the age of 18 month and the polyunsaturated fatty acid in muscle along with the PUFA/SFA ratio reached the highest levels at this age. Inosine 5′-monophosphate (IMP) was the predominant umami compound in beef, which concentration was significantly higher (*p* < 0.05) at month 18, but not different between Angus and Xiangxi yellow cattle. Multiple volatile flavor compounds were higher (*p* < 0.05) in concentrations in meat from Xiangxi yellow cattle at ages of 18 and 30 months when compared to Angus. Sensory analysis revealed that Xiangxi yellow cattle (18 and 30 months) and Angus (30 months) were superior in meat overall eating quality to Xiangxi yellow cattle (6 months) and Angus (6 and 18 months). This study showed that Xiangxi yellow cattle are a fine cattle breed with equal or even better meat quality attributes when compared to Angus. It is proper to slaughter Xiangxi yellow cattle at the age of 18 months for high quality beef production.

## 1. Introduction

Meat from different breeds of animals differs in quantitative and qualitative (compositional and sensory) [1,2,3]. The meat qualitative variations may be attributed to genetic factors, physiological stages, muscles types, sex, and ages of animals as internal factors while a list of external factors including feed differences also have impact [4,5,6]. Aberdeen Angus is the world-renowned beef cattle breed, it can attain high marbling levels, especially on a high nutritional plane [7], and has the characteristics of good eat quality [8]. Xiangxi yellow cattle are generally small in body size and the mature weight usually less than 400 kg. It is an indigenous breed from the northwest of Hunan province, China. With the agricultural mechanization and the increasing demand for meat, Xiangxi yellow cattle are increasingly being raised for meat production. Xiangxi cattle is the optimal meat for consumption with better quality, better taste and higher nutritional value than other ordinary beef breeds of China [9,10].

The dressing percentage and lean meat yield of Xiangxi yellow cattle are 52.3 and 40.8%, respectively [11]. For this reason, previous studies on Xiangxi yellow cattle were mostly on the growth performance and efforts have been made to improve meat yield per head, such as crossing superior foreign breads with Xiangxi yellow cattle and nutrition manipulation [12,13]. People’s demand for beef has entered the era of quality and quantity coexistence, the demand for high-grade beef is strong, there is an urgent need to improve the output of high-grade beef. However, the meat properties of Xiangxi yellow cattle were not well studied at the same time. Little is known about the changes in muscle composition and eating quality of the cattle at different ages. In the present study, the muscle proximate composition, meat flavor substances and sensory quality of Xiangxi yellow cattle at different growth stages were profiled in comparison to Aberdeen Angus cattle, which may provide valuable information for high grade beef production from Xiangxi yellow cattle. The valuable information may benefit us in further explore the local varieties that can be used in the production of high-grade beef to reduce the gap of high-grade beef products and provide scientific basis for the cultivation and innovation of high-grade beef cattle.

## 2. Materials and Methods

### 2.1. Animals and Muscle Sampling

All animals used in this study were fed on the farm of Hunan Denong Animal Husbandry Technology Co., Ltd. (Huayuan, China). After weaned at the age of 4 months, animals were fed concentrates on daily basis at the rate of 0.5% of body weight. The concentrate ingredients were corn, soybean meal, wheat bran, premix, salt and calcium bicarbonate in a proportion as 45:30:19.5:4:0.5:1 at the ages of 6 months. The concentrate ingredients were corn, soybean meal, wheat bran, premix, salt and calcium bicarbonate in a proportion as 50:26:18.5:4:0.5:1 at the ages of 18 months. The concentrate ingredients were corn, soybean meal, wheat bran, premix, salt, baking soda and calcium bicarbonate in a proportion as 55:24:15:4:0.5:0.5:1.0 at the ages of 30 months. Different breeds of cattle at the same age were fed the same diet. The cattle were fed on forage and the concentrate was offered at 07:00 a.m. and 02:30 p.m. every day with free access to water all of the time. All animals were healthy and the immunization procedures were consistent and unified.

At the ages of 6, 18, and 30 months, 10 female animals for both Xiangxi yellow and Aberdeen Angus cattle were randomly selected and slaughtered at a commercial slaughter plant (Hunan Denong Animal Husbandry Technology Co., Ltd., Huayuan, China) according to the commercial procedure. Animals were kept off-fed for 24 h before slaughter. After slaughter, carcasses were aged at 0–4 °C for 3 days. A slice of Longissimus thoracis (LT) muscle between the 12th and 13th ribs were dissected, quickly cut into small pieces, mixed, snapfrozen in liquid nitrogen and stored at −80 °C for proximate composition, free amino acids (FAAs), 5′-nucleotides, fatty acids and volatile compounds analysis. At the same time, one boneless loin chop was dissected from the same location and used for sensory evaluation.

### 2.2. Proximate Composition Analysis

Lipid, moisture, protein and ash in LT muscle were determined according to GB/T 5009.6-2016 (national standard for lipid analysis), GB/T 5009.3-2016 (national standard to analyze moisture), GB/T 5009.5-2016 (national standard for protein) and national standard for ash analysis (GB/T 5009.4-2016) in foods, respectively.

### 2.3. Free Amino Acid Analysis

The free amino acids (FAAs) in muscle were analyzed by HPLC as previously described [14,15] with some modification. Briefly, 0.5 g of cooked meat (100 °C, 5 min) was homogenized using a ULTRA TURRAX disperser (IKA, Staufen, Germany) in 5 mL of pure water for 5 min. After centrifugation at 12,000× *g*, 4 °C for 15 min, the supernatant was collected. The pellet was added with 2 mL of pure water and extraction was repeated. The supernatants were combined, added with 1 mL of 30% (*m*/*v*) zinc acetate, and made to 10 mL with H_2_O. After centrifugation at 12,000× *g* for 15 min, the supernatant was passed through a 0.22 µm filter before the amino acids were derivatized.

Pre-column derivatization was performed using ortho-phthalaldehyde (OPA) for the primary amino acids and 9-fluorenylmethyloxycarbonyl (FMOC) for the secondary amino acids in 0.4 M borate buffer (pH 10.2). Samples were analyzed as previously described [14] using an Agilent 1100 HPLC (Agilent Technology Inc., Waldbronn, Germany) equipped with a UV DAD detector. In the present study, a mixed amino acid standard (Sigma) contain 17 amino acids (Asp, Glu, Ser, His, Gly, Thr, Arg, Ala, Tyr, Cys, Va1, Met, Phe, Ile, Leu, Lys, and Pro), Asn, Gln, Cit, Nva, Trp, Hyp, and Sar were used as standards for quantification [14,15]. 

### 2.4. Analysis of 5′-Nucleotides

Nucleotides in muscle, including AMP, IMP and GMP, were analyzed by HPLC as previously described [16]. Briefly, 0.2 g of LT muscle was cooked (100 °C, 5 min) and homogenized in 0.8 mL of ice-cold 5% perchloric acid. The homogenate was centrifuged at 10,000× *g*, 4 °C for 10 min and the supernatant was collected. The pellet was homogenized again in the same volume of 5% perchloric acid. Supernatant from both homogenization was combined, neutralized to pH 5.93, centrifuged to remove KClO4, and ultra-filtrated through a 0.22 µm filter before injected into a Waters Model 2695 separations LC system (Waters, Milford, MA, USA). Chromatographic separation, identification and quantification of nucleotides in samples were performed as in our previous study [16].

### 2.5. Fatty Acid Analysis

Fatty acids in LT muscle were determined according to the procedure of Yu et al. [17]. Briefly, 5 g of minced muscle was freeze-dried and powdered. 0.5 g of powdered muscle was added with 1.0 mL of 1.0 mg/mL glycerol triundecanoate (Nu-Chek Prep Inc., Minnesota, PA, USA) in methanol as internal standard and 4 mL of benzene: petroleum ether (1:1) for the extraction of total lipid. The lipid extract was added with 4 mL of 0.4 M KOH in methanol and vortexed to prepare fatty acid methyl esters (FAME). Post methylation, samples were added with 10 mL of saturated sodium chloride solution, vortexed and allowed to separate into layers. The top layer containing FAME was collected, added with 1.0 g of Na_2_SO_4_, and centrifuged at 800× *g* for 5 min. 0.1 mL of the supernatant containing FAME was diluted to 1 mL with hexane and transferred into a 2-mL glass vial before GC-MS analysis.

The FAME were separated on a DB-5MS column (length 30 m, internal diameter 0.25 mm, 0.25 µm film thickness; Agilent Technologies Co., Palo Alto, CA, USA) using a GCMS-QP2010 gas chromatograph mass spectrometer (Shimadzu, Tokyo, Japan). For each sample, 2 μL of prepared FAME was injected and the injection temperature was set at 250 °C. The injector was operated in split mode with a split ratio of 10:1. The carrier gas was helium and the flow rate was 1 mL/min. The column oven temperature was held at 50 °C for 1 min, increased to 160 °C at 20 °C/min, held at 160 °C for 1 min, increased to 250 °C at 10 °C/min and finally held at 250 °C for 10 min. The GC/MS interface was heated at 250 °C. The acquisition of mass spectra was performed in electron impact (EI) mode (70 eV) with full scan, scanning the mass range *m*/*z* 33–550. A mixed standard containing 37 FAME (Supelco Inc., Bellefonte, PA, USA) was used as external standards for peak identification and quantification. The FAME which was not present in the mixed standard were identified and quantified by comparing mass spectra available in National Institute of Standards and Technology (NIST 11 and NIST 11s) and the peak area of internal standard, respectively.

### 2.6. Volatile Compounds Analysis

Volatile compounds were analyzed by GC-MS. The headspace volatile compounds were extracted by solid-phase micro-extraction (SPME) according to the procedure of Gabriel et al. [16] using 2-methyl-3-heptanone (Sigma, St. Louis, MO, USA) as internal standard. GC-MS analysis was performed using a GCMS-QP2010 gas chromatograph mass spectrometer (Shimadzu, Tokyo, Japan). The injector, operated in split-less mode, was set at 250 °C with 4 min desorption time. Helium was used as carrier gas at flow rate 1.7 mL/min. Volatile compounds were separated on a Rtx-5MS column (length 30 m, internal diameter 0.25 mm, film thickness 0.25 µm; Shimadzu). The temperature of column oven was kept at 40 °C for 3 min, increased to 70 °C at 4 °C/min and then to 230 °C at 3 °C/min, and held at 230 °C for 5 min. The GC/MS interface was heated at 240 °C. Mass spectra were acquired in electron impact (EI) mode (70 eV) at 10 microscan/s, scanning the mass range *m*/*z* 25–300. Compounds were identified by the mass spectra and linear retention indexes (LRI) as previously described [16].

### 2.7. Sensory Evaluation

The sensory analysis of meat was carried out by a 10-persoon trained taste panel [16]. Boneless loin chops (2 cm thickness) were broiled in pan to an internal temperature of 74 °C, cut into 2 cm^3^ cubes, wrapped in pre-labeled foils and placed in a heated incubator before given to the assessors. Six samples (2 breeds × 3 ages) were provided to each panelist in a session. Testing samples were scored on a 1–8 point scale for parameters of tenderness (1 = extremely tough and 8 = extremely tender), juiciness (1 = extremely dry and 8 = extremely juicy), flavor liking and overall liking (1 = extremely disliking and 8 = extremely liking), and beefy flavor and abnormal flavor (1 = extremely weak and 8 = extremely strong).

### 2.8. Statistical Analysis

Data were analyzed by using SPSS (version 17.0, SPSS Inc., Chicago, IL, USA). The data among 6 groups were analyzed by one-way analysis of variance followed by Fisher’s protected least significant difference test Breed, month and their interactions were analyzed by mixed model procedure. Differences were considered significant at *p* < 0.05. 

## 3. Results and Discussion

### 3.1. Proximate Composition

Proximate composition of LT muscle was listed in Figure 1. The moisture content in muscle from Xiangxi yellow cattle at any ages was significantly higher (*p* < 0.05) than that of 30 months Angus. Intercellular moisture plays an important role in muscle tenderness, and the more water a muscle can hold, the better its tenderness [18], indicating that xiangxi yellow cattle muscle contains more water, tenderness may be higher than Aberdeen Angus at the same age. The lipid content was higher (*p* < 0.05) in muscle from Xiangxi yellow cattle at the age of 18 months, but not different (*p* > 0.05) between the two breeds at other ages. As intramuscular fat or marbling is a very important factor for meat quality and meat grading, these data indicate that, Xiangxi yellow cattle may is a fine breed for high quality beef production same as Angus. It also showed that Xiangxi yellow cattle is an early maturing breed [11] as the intramuscular fat content at 18 month reached the same level as at 30 month. Ash content in muscle did not change with ages and no difference was determined between the two breeds at all three ages (Figure 1), but protein content was generally lower in the muscle from Xiangxi yellow cattle, especially at the age of 30 months the difference was significant (*p* < 0.05). In summary, Xiangxi yellow cattle may be an excellent breed for high quality beef production based on the intramuscular fat and moisture content.

### 3.2. Free Amino Acids

Beef provides a wide range of essential nutrients, in particular digestible proteins of high biological value. The digestibility of beef protein at the end of small intestine was very high (90–95%), as measured in humans by Oberli et al. [19,20]. Proteins break down to produce free amino acids, certain free amino acids (e.g., glutamate, glycine and β-alanine) provide “meaty flavor” to improve appetite and gastrointestinal function [21]. Free amino acids (FAAs) due to their specific taste have a great importance role in food eating quality [22]. There were 24 FAAs determined in LT muscle from Angus and Xiangxi yellow cattle. The 24 FAAs taste attributes, concentrations and taste threshold of FAAs were listed in Table 1. The content of most FAAs was under 10 mg/100 g muscle. Alanine was the most abundant free amino acid in Xiangxi yellow cattle and Aberdeen Angus muscle, which reached 90.65–156.12 mg/100 g muscle. The concentrations of free alanine were lower in muscle from Xiangxi yellow cattle at different age when compared to Aberdeen Angus. Besides, age had no significant effect on the concentration of free alanine in muscle between the two cattle breeds. (Table 1). In addition, alanine was the only free amino acid with taste activity value (TAV) being greater than 1. As a compound with TAV < 1 has less contribution to the taste and the compounds with TAV > 1 are considered as contributors to taste [23]. These data revealed that alanine maybe a contributor to beef taste and the difference in alanine concentrations (Table 1) may contribute to difference in meat taste between Angus and Xiangxi yellow cattle. The total free amino acids were significantly higher (*p* < 0.05) in muscle from Angus than in muscle from Xiangxi yellow cattle. In addition to alanine, multiple free amino acids in muscle were determined to be significantly different (*p* < 0.05) between the two breeds, which included Asn, Gly, Cys, Val, Met, norvaline (Nva), Phe, Leu, and the subtotal umami and sweet amino acids. The TAV of the above free amino acids with significant differences among varieties were all less than 1, thus, these amino acids should not induce difference in meat taste between the two cattle breeds. The ages of cattle showed much less effect on free amino acids in muscle, with only Cys and sarcosine (Sar) being significantly influenced by the ages of animals (Table 1). This is in agreement with previous study on pigs which reports that genetic factors have a major influence on FAA in pork [24]. However, some literatures report that the total amino acids in bovine muscle are muscle type specific, which are different between different muscles, but for the same muscle, total amino acids are similar across cattle breeds [25,26,27]. Proteolysis is responsible for the postmortem aging of meat. The difference of free amino acids between Angus and Xiangxi yellow cattle could be induced by the different protease activities in postmortem muscle [28].

### 3.3. 5’-nucleotides

The 5′-nucleotides are important components of umami taste which is caused by the interaction of glutamate with tongue receptors [29,30]. Similar as in pork [16,31,32], IMP is the dominant umami 5’-nucleotide in beef, which concentration was much higher than those of GMP and AMP (Table 2). To evaluate the umami taste of beef, the equivalent umami concentration (EUC) and the TAV of different samples were calculated [23]. The TAV of IMP were in the range of 4.62–7.73, greater than 1. However, the TAV of any other umami compounds, including AMP, GMP and umami free amino acids (Table 1) were all <1. These data revealed that IMP was the major umami source of beef. Results from this study revealed that both breeds and age at slaughter had significant effects on the concentration of IMP in meat. That is to say, the umami taste of beef was influenced by both cattle breed and ages. For both Angus and Xiangxi yellow cattle, the IMP concentrations were higher in muscle at 18 months (Table 1). EUC refers to the concentration of monosodium glutamate (MSG) that is equivalent to the umami intensity given by the mixture of MSG-like amino acids (aspartic acid and glutamic acid) and 5-nucleotides (GMP, IMP, and AMP). Yamaguchi et al. [33] proposed to use EUC value to represent umami substance content in samples and objectively evaluate the freshening effect of food. Similar as IMP concentrations in meat, the calculated EUC and the TAV of EUC was significantly influenced by cattle breeds. For Xiangxi yellow cattle, the EUC at 18 months were higher than at month 6 and 30 months, but which no significantly different when compared with Angus at 18 months. In summary, these data showed that IMP was the primary umami source of beef. The EUC from Xiangxi yellow cattle reached highest values at 18 moths, which were not different when compared with Angus at 18 months.

### 3.4. Fatty Acid Content

Fatty acid content and composition in meat are not only important to human health, but also influence meat taste and formation of volatile compounds [34]. In the present study, 22 fatty acids in total were determined in the LT muscle from Angus and Xiangxi yellow cattle, including 10 saturated fatty acids (SFA), 6 monounsaturated fatty acids (MUFA) and 6 polyunsaturated fatty acids (PUFA) (Table 3). The content of most of these fatty acids was influence by cattle breed, ages or both except C18: 1n9t and C22: 6n3. As previously reported [35], C18: 1n9c was the most abundant fatty acid in beef, followed by C16: 0 and C18: 0, which concentration were all over 100 mg/100 g muscle. 

Most fatty acids determined in the present study increased with the increasing ages of cattle, which led to the SFA, MUFA and total fatty acids were significantly higher (*p* < 0.05) in muscle at 30 months for both Angus and Xiangxi yellow cattle, but there were not different between the two breeds. This is in agreement with lipid content in muscle (Figure 1), showing increased body fatty deposition with the growth of animals. However, PUFA in muscle from Xiangxi yellow cattle reached the highest (153.03 mg/100 g muscle, *p* < 0.05) level at 18 months and decreased (*p* < 0.05) afterwards. In fact, PUFA in muscle from Angus also increased significantly from 6 months (83.73 mg/100 g muscle) to months (141.93 mg/100 g muscle) and numerically decreased at 30 months. (120.84 mg/100 g muscle) though the decrease was not statistically different (*p* > 0.05). The results show that the different deposition pattern of this subgroup of fatty acids between the two breeds. The PUFA/SFA ratios was higher at 18 months for both Angus and Xiangxi yellow cattle, which closer to the minimum ratio of polyunsaturated to saturated fatty acids recommended by nutritionists [36]. In addition, the concentrations of n − 3 and n − 6 PUFA were higher or highest (*p* < 0.05) at 18 months for both Angust and Xiangxi yellow cattle. When the two breeds were compared, the content of MUFA and total fatty acids were higher (*p* < 0.05) in muscle from Xiangxi yellow cattle at 18 months than in muscle from Angus. Based on the content of lipid (Figure 1) and fatty acids (Table 3) in muscle, Xiangxi yellow cattle can be slaughtered at age of 18 months for high quality beef production.

### 3.5. Volatile Compounds

The volatile flavor compounds (44 in total) were detected and quantified in LT muscle of cattle (Table 4). As previously reported [37], these compounds can be divided into eight groups: alcohols (6), aldehydes (12), ketones (2), organic acids (4), esters (2), hydrocarbons (14), heterocyclic compounds (2) and others (2). Same as previously reported in beef [35] and pork [16,38], aldehydes were the most predominant class of volatile compounds with concentrations in the range of 43,748.53–72,511.41 ng/100 g muscle. Although two more types of compounds were identified, the subtotal concentration of hydrocarbons was lower than that of aldehydes (Table 4). The subtotal concentrations of alcohol, ketones, organic aicds and esters were all lower than the concentration of hydrocarbons, but they should contribute more to meat flavor as their odor-detection thresholds were much lower and hydrocarbons are generally considered to contribute little to meat flavor. Among the 44 identified volatile compounds, 26 were affected by cattle breeds, ages or both, but more hydrocarbons (8 vs. 6) were not varied between any muscle samples. 

Aldehydes are important to meat aromas as their odor-detection thresholds are low [39]. In the present study, several aldehydes were determined to be increased in meat with animal growth. These aldehydes were dodecanal, tetradecanal, and hexadecanal. As aldehydes are primarily generated from thermal oxidation of fatty acids during cooking, especially C18:2n6 and C18:3n3 in meat [40,41], the increased amount of these compounds in meat from older animal could be explained by the increased concentrations of these two fatty acids in muscle (Table 3). In addition, tridecenal, tetradecanal, pentadecanal, hexadecanal, heptadecanal, and octadecanal were detected to be higher in meat from Xiangxi yellow cattle at all ages, which may be related to the higher concentration of C18:3n3 in muscle from Xiangxi yellow cattle, especially the higher content of this fatty acid at 18 and 30 months (Table 3). The subtotal concentrations of aldehydes were significantly higher in meat from Xiangxi yellow cattle at all three ages, indicating genetic factors had a major effect on this class of flavor compounds in beef.

Six alcohols were detected in the present study (Table 4), among which dodecanol, hexadecanol, and 2-hexyl-1-decanol were not varied in amount in meat, indicating that these compounds were not impacted by cattle breed or ages. When the two breeds were compared, the content of octanol, 1-octen-3-ol, 2-octyl-1-decanol and thus the subtotal concentration of alcohols were significantly higher (*p* < 0.05) in meat of Xiangxi yellow cattle at all ages. As C18:3n3 is believed to be the source of 1-Octen-3-ol due to its third double bond [35], the much higher (*p* < 0.05) concentration of this compound in meat from Xiangxi yellow cattle at month 18 and 30 could be related to the higher concentration of this polyunsaturated fatty acid in muscle ((Table 3). 

Ketones are also formed from fatty acid oxidation [42]. In the present study, only two ketones, 2-Undecanone and 6,10-Ddimethylundeca-5,9-Dien-2-One, were identified and quantified in beef (Table 4). 2-undecanone, which has citrus oil and rutin-like aroma, was determined to be significantly higher (*p* < 0.05) in muscle from Xiangxi yellow cattle at month 18 and 30 than in any other muscle samples. 

As previously reported [35], hexanoic acid and nonanoic acid were detected in beef in the present study. These two organic acids were higher in concentrations in meat of Xiangxi yellow cattle (Table 4). In addition, two more organic acids, tetradecanoic acid and hexadecanoic acid, were determined, but the concentration of tetradecanoic acid was not impacted by either cattle breeds or ages. Esters usually have low odor-detection threshold and are believed to be important flavor compounds of fermented meat products [43]. In the present study, the typical ester compounds of fermented meat products, ethyl acetate and ethyl lactate, were not detected in fresh beef, but two other esters, sulfurous acid 2-ethylhexyl isohexyl ester and diisobutyl phthalate were detected (Table 4). As these two compounds are not natural meat ingredient and diisobutyl phthalate is a commonly used plasticizer, they should be contaminants from environment and feed. Their difference in concentrations may reflect the different metabolism and deposition of the two compounds within body between Angus and Xiangxi yellow cattle. Same as in literature, hydrocarbons were the second most abundant volatile compounds following aldehydes in meat [35,38]. Among the 14 hydrocarbons identified in the present study, 8 hydrocarbons did not change in concentration between muscle samples, but the other 6 hydrocarbons, were significantly higher (*p* < 0.05) in the muscle from Xiangxi yellow cattle at month 18 and 30 than in other muscle samples (Table 4). As hydrocarbons have high aroma threshold, they usually contribute trivial to cooked meat flavor. In summary, the meat of Xiangxi yellow cattle at 18 and 30 months had higher concentrations of multiple volatile flavor compounds, which included alcohols (octanol, 1-octen-3-ol, and 2-octyl-1-decanol), aldehydes (tridecenal, tetradecanal, pentadecanal, hexadecanal, heptadecanal, and octadecanal), ketones (2-undecanone) and hydrocarbons 

### 3.6. Sensory Quality

The sensory quality of beef was evaluated by a trained taste panel and the results are listed in Table 5. Meat tenderness was significantly lower (*p* < 0.05) for both Angus and Xiangxi yellow cattle at month 30 than at months 6 and 18, but the two breeds were not different at the same sampling time points. This is logical since meat tenderness decreases with the maturation of animals [44,45]. Statistical analysis showed that cattle breeds had no significant influence on meat juiciness, but months did impact on meat juiciness. For both Angus and Xiangxi yellow cattle, the meat juiciness score of 18 and 30 months was significantly higher than that at 6 months (*p* < 0.05). This should be explained by the increased intramuscular fat deposition (Figure 1). Meat juiciness was not different between Angus and Xiangxi yellow cattle at the same slaughter ages. It has been previously reported that increased intramuscular fat increases beef flavor intensity [46]. In agreement with literature, the beefy flavor of meat from both Angus and Xiangxi yellow cattle increased at 18 months. In addition, both beefy and abnormal flavors of meat from Xiangxi yellow cattle at months were significantly higher (*p* < 0.05) than those of Angus at any ages. This should be related to the increased concentrations of multiple volatile flavor compounds in meat from Xiangxi yellow cattle (Table 4). However, the flavor liking and overall liking scores were not different between meat samples.

To better understand the impact of breeds and slaughter ages on the eating quality of beef, principal component analysis (PCA) was performed. As shown in Figure 2, PCA1 explained 48.55% and PC2 explained 28.25% of the variance associated with meat sensory attributes. PC1 could be explained to represent the “overall eating quality” of beef, which separated Xiangxi yellow cattle (18 and 30 months) and Angus (30 months) from six treatments groups. Showing that the treatments group separated by PC1 had better overall eating quality than the rest treatments group. PC2 separated Xiangxi yellow cattle from Angus, showing that genetic factors had an important impact on beef sensory attributes. Xiangxi yellow cattle (18 months) located close to beefy flavor in the same quadrant, suggests that the high overall eating quality of this treatment maybe was related to its increased intensity of beefy flavor. However, the intensity of beefy flavor may- was associated to the increased content of multiple volatile flavor compounds.

## 4. Conclusions

Muscle proximate composition, fatty acid profile, meat flavor compounds and sensory quality of Xiangxi yellow cattle slaughtered at ages of 6, 18 and 30 months were analyzed in comparison to Aberdeen Angus. The results of this study, the IMF, IMP, PUFA, PUFA/SFA ratio and Multiple flavor compounds in LT muscle, which at 18 months higher than at 6 months on Xiangxi yellow cattle. Xiangxi cattle aged 18 months and 30 months had the same IMF in LT muscle. The content of PUFA and the PUFA/SFA ratio maximized at age of 18 months in muscle from Xiangxi yellow cattle. There was no significant difference in IMF and IMP concentrations between 18-months Xiangxi cattle and Angus cattle. Multiple flavor compounds were higher in concentrations in meat from Xiangxi yellow cattle at ages of 18 and 30 months when compared to Angus. Meat juiciness, tenderness, flavor liking and overall liking scores were not different between Angus and Xiangxi yellow cattle at the same slaughter ages. As the result, xiangxi yellow cattle is a fine cattle breed with equal or even better meat quality attributes when compared to Angus. It is proper to slaughter Xiangxi yellow cattle at the age of 18 months for high quality beef production. The finding of the present study may provide valuable information for high grade beef production from Xiangxi yellow cattle and provide scientific basis for the cultivation and innovation of high-grade beef cattle.

## Figures and Tables

**Figure 1 animals-12-01161-f001:**
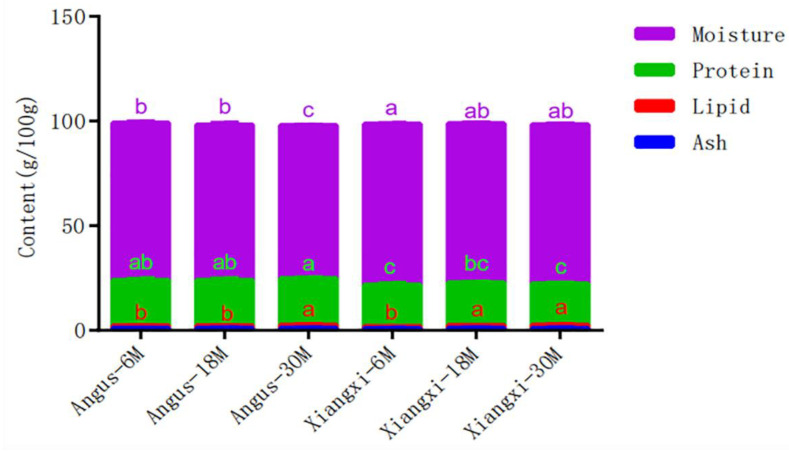
Proximate composition (g/100 g muscle) of LT muscle from Angus and Xiangxi yellow ca ttle. For the same ingredient, means lacking a common letter are significantly different (*p* < 0.05).

**Figure 2 animals-12-01161-f002:**
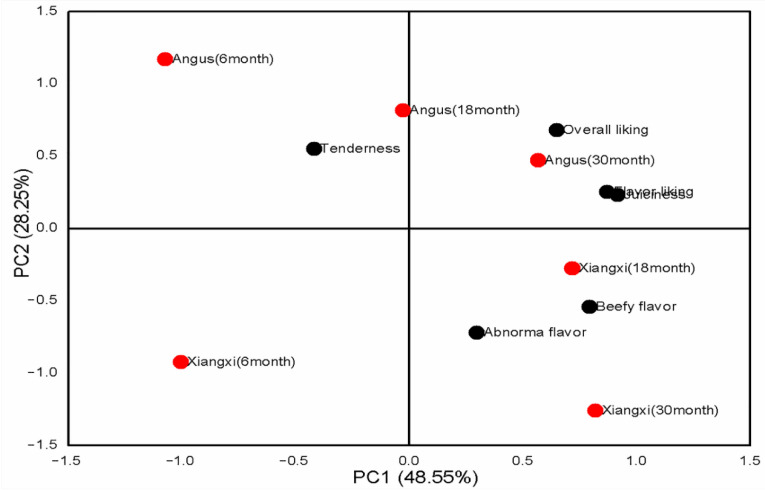
Principle component analysis (PCA) for sensory attributes of LT muscle from Angus and Xiangxi yellow cattle.

**Table 1 animals-12-01161-t001:** Free amino acid concentrations (mg/100 g) determined in LT muscle from Angus and Xiangxi yellow cattle.

Free Amino Acids	Threshold (mg/100 g)	Taste Attribute	Angus			Xiangxi Yellow Cattle			s.e.m.	Sig.		
			6 Month	18 Month	30 Month	6 Month	18 Month	30 Month		Breed	Month	B × M
Asp	100	Uma (+)	0.98	0.98	0.78	N.D.	1.53	1.43	0.16	—	—	—.
Glu	30	Uma (+)	5.45	5.52	5.88	5.77	4.58	3.70	0.32	n.s.	n.s.	n.s.
Asn	100	Uma (+)	3.13 ^abc^	3.93 ^ab^	4.68 ^a^	2.02 ^bc^	1.23 ^c^	1.69 ^c^	0.41	**	n.s.	n.s.
Ser	150	Swt (+)	3.94	5.34	4.38	5.06	4.23	5.50	0.24	n.s.	n.s.	n.s.
Gln	—	—	30.97	45.04	51.81	33.56	38.09	40.03	2.62	n.s.	n.s.	n.s.
His	20	Bit (−)	1.24	1.80	0.86	1.60	N.D.	N.D.	0.22	—	—	—
Gly	130	Swt (+)	11.84 ^b^	17.21 ^ab^	11.50 ^b^	23.37 ^a^	18.12 ^ab^	19.38 ^a^	1.38	**	n.s.	n.s.
Thr	260	Swt (+)	3.59	4.38	3.20	3.70	3.02	3.70	0.22	n.s.	n.s.	n.s.
Cit	—	—	N.D.	N.D.	N.D.	0.63	N.D.	N.D.	0.07	—	—	—
Arg	50	Bit/Swt (+)	5.39	5.26	5.03	4.80	5.09	7.91	0.43	n.s.	n.s.	n.s.
Ala	60	Swt (+)	148.23 ^ab^	156.12 ^a^	136.36 ^ab^	96.85 ^b^	115.83 ^ab^	90.65 ^b^	8.87	*	n.s.	n.s.
Tyr	—	Bit (−)	2.56	4.56	3.65	3.90	3.32	2.47	0.36	n.s.	n.s.	n.s.
Cys	—	Bit/Swt/Sul (-)	2.09 ^d^	2.74 ^cd^	2.27 ^d^	3.32 ^c^	4.21 ^b^	5.49 ^a^	0.37	***	**	*
Val	40	Swt/Bit (−)	6.05 ^a^	7.01 ^a^	5.72 ^a^	2.75 ^b^	2.82 ^b^	4.53 ^ab^	0.53	**	n.s.	n.s.
Met	30	Bit/Swt/Sul (-)	3.45 ^ab^	4.70 ^a^	3.59 ^ab^	1.84 ^bc^	2.02 ^bc^	1.37 ^c^	0.39	**	n.s.	n.s.
Nva	—	—	48.26 ^a^	47.71 ^a^	38.56 ^a^	10.43 ^b^	10.87 ^b^	9.29 ^b^	5.51	***	n.s.	n.s.
Trp	—	—	10.80	5.29	8.54	N.D.	N.D.	N.D.	1.33	—	—	—
Phe	90	Bit (−)	9.70 ^b^	10.81 ^ab^	9.65 ^b^	12.93 ^a^	11.84 ^ab^	11.23 ^ab^	0.44	*	n.s.	n.s.
Ile	90	Bit (−)	6.60	7.87	6.85	6.86	5.95	6.21	0.36	n.s.	n.s.	n.s.
Leu	190	Bit (−)	8.68 ^a^	10.04 ^a^	8.82 ^a^	4.15 ^b^	3.89 ^b^	3.65 ^b^	0.85	***	n.s.	n.s.
Lys	50	Swt/Bit (−)	3.69	4.62	2.75	2.21	2.39	2.23	0.35	n.s.	n.s.	n.s.
Hyp	—	—	0.84	1.02	0.82	1.02	0.56	1.03	0.08	n.s.	n.s.	n.s.
Sar	—	—	0.69 ^b^	0.50 ^b^	1.89 ^a^	0.56 ^b^	0.79 ^b^	1.92 ^a^	0.18	n.s.	***	n.s.
Pro	300	Swt/Bit (−)	1.78	2.21	1.17	2.58	1.14	1.23	0.20	n.s.	n.s.	n.s.
Umami AA	—	—	9.56 ^b^	10.43 ^ab^	11.34 ^a^	7.79 ^c^	7.34 ^c^	6.82 ^c^	0.52	***	n.s.	n.s.
Sweet AA	—	—	179.10 ^ab^	196.88 ^a^	165.07 ^ab^	136.51 ^b^	147.54 ^ab^	127.21 ^b^	8.80	**	n.s.	n.s.
Bitter AA	—	—	39.69	47.77	40.71	39.38	36.31	38.32	1.61	n.s.	n.s.	n.s.
Total	—	—	319.90 ^a^	354.61 ^a^	318.71 ^a^	236.86 ^b^	241.47 ^b^	224.60 ^b^	16.23	**	n.s.	n.s.

N.D: not detected. ^abcd^ Values in the same row without a common letter are significantly different (*p* < 0.05) Sig.: significance; n.s.: not significant; * *p* < 0.05; ** *p* < 0.01; *** *p* < 0.001.

**Table 2 animals-12-01161-t002:** The concentrations of 5′-nucleotides (μg/g), the calculated EUC (g MSG/100 g) and TAV of LT muscle from Angus and Xiangxi yellow cattle.

	Threshold (μg/g)	Angus			Xiangxi Yellow Cattle			s.e.m.	Sig.		
		6 Month	18 Month	30 Month	6 Month	18 Month	30 Month		Breed	Month	B × M
The concentrations of 5′-nucleotide and EUC											
5′-GMP	125	56.19 ^b^	50.86 ^b^	50.46 ^b^	34.09 ^c^	60.95 ^b^	71.88 ^a^	3.57	n.s.	**	**
5′-IMP	250	1636.27 ^bc^	1901.33 ^a^	1810.32 ^ab^	1155.67 ^d^	1930.63 ^a^	1566.70 ^c^	81.19	**	***	*
5′-AMP	500	104.46 ^a^	89.50 ^b^	87.96 ^b^	77.96 ^b^	90.82 ^b^	105.08 ^a^	3.09	n.s.	n.s.	**
EUC	300	1.20 ^ab^	1.39 ^a^	1.42 ^a^	0.89 ^b^	1.20 ^ab^	0.82 ^b^	0.09	*	n.s.	n.s.
TAV of 5′-Nucleotide and EUC											
5′-GMP	125	0.45 ^b^	0.41 ^b^	0.41 ^b^	0.28 ^c^	0.49 ^b^	0.58 ^a^	0.03	*	***	***
5′-IMP	250	6.55 ^bc^	7.61 ^a^	7.24 ^ab^	4.62 ^d^	7.73 ^a^	6.27 ^c^	0.33	**	***	**
5′-AMP	500	0.21 ^a^	0.18 ^ab^	0.17 ^b^	0.16 ^b^	0.18 ^ab^	0.21 ^a^	0.01	n.s.	n.s.	**
EUC	300	39.98 ^ab^	46.25 ^a^	47.24 ^a^	29.46 ^b^	39.87 ^ab^	27.26 ^b^	2.86	*	n.s.	n.s.

TAV, taste activity value, is the ratio of a compound concentration to its taste threshold. Equivalent umami concentration (EUC) is the concentration of monosodium glutamate (MSG) that is equivalent to the umami intensity given by the mixture of MSG-like amino acids (aspartic acid and glutamic acid) and 5′-nucleotides (GMP, IMP and AMP). ^abcd^ Values in the same row without a common letter are significantly different (*p* < 0.05). Sig.: significance; n.s.: not significant; * *p* < 0.05; ** *p* < 0.01; *** *p* < 0.001.

**Table 3 animals-12-01161-t003:** Fatty acid concentrations (mg/100 g muscle) in the total lipid fraction of intramuscular fat of LT muscle from Angus and Xiangxi yellow cattle.

Fatty Acids	Angus			Xiangxi Yellow Cattle			s.e.m.	Sig.		
	6 Month	18 Month	30 Month	6 Month	18 Month	30 Month		Breed	Month	B × M
C10: 0	1.85 ^bc^	2.75 ^a^	3.01 ^a^	1.58 ^c^	2.87 ^a^	2.12 ^b^	0.17	**	***	**
C12: 0	1.73 ^e^	2.94 ^c^	3.54 ^b^	2.32 ^d^	5.36 ^a^	2.55 ^d^	0.35	***	***	***
C13: 0	0.37	0.22	0.49	N.D.	0.79	0.38	0.12	—	—	—
C14: 0	9.20 ^d^	21.13 ^c^	34.93 ^a^	7.67 ^d^	28.52 ^b^	34.04 ^a^	3.32	n.s.	***	**
C14: 1	4.42 ^b^	6.27 ^b^	6.00 ^b^	11.23 ^a^	9.50 ^a^	8.87 ^a^	0.73	***	n.s.	n.s.
C15: 0	6.52 ^d^	11.67 ^c^	10.93 ^c^	13.96 ^b^	15.02 ^b^	22.26 ^a^	1.45	***	***	***
C16: 0	156.37 ^d^	193.69 ^bc^	215.25 ^ab^	166.03 ^cd^	166.83 ^cd^	225.73 ^a^	8.24	n.s.	**	n.s.
C16: 1	3.08 ^c^	3.42 ^c^	3.68 ^c^	4.66 ^bc^	8.18 ^a^	5.59 ^b^	0.54	***	*	*
C17: 0	24.63 ^c^	48.16 ^b^	42.83 ^b^	40.58 ^b^	43.91 ^b^	67.91 ^a^	3.91	**	***	**
C17: 1	7.51 ^de^	9.48 ^cd^	10.56 ^bc^	6.33 ^e^	11.67 ^b^	14.98 ^a^	0.87	**	***	**
C18: 0	157.92 ^bc^	163.89 ^bc^	185.35 ^ab^	117.63 ^d^	136.18 ^cd^	200.65 ^a^	8.82	*	**	*
C18: 1 n9 t	1.24	0.96	1.13	1.59	1.45	1.67	0.10	n.s.	n.s.	n.s.
C18: 1 n9 c	313.13 ^c^	318.21 ^c^	685.23 ^a^	233.24 ^c^	497.92 ^b^	675.58 ^a^	54.41	n.s.	***	**
C18: 2 n6	38.41 ^b^	87.66 ^a^	72.87 ^a^	53.37 ^b^	74.86 ^a^	52.12 ^b^	5.17	n.s.	***	*
C18: 3 n3	4.42 ^d^	5.20 ^d^	6.19 ^bc^	4.92 ^d^	10.64 ^ab^	13.63 ^a^	1.11	**	*	n.s.
C19: 0	0.93 ^d^	1.91 ^c^	4.79 ^a^	3.25 ^b^	3.89 ^b^	4.68 ^a^	0.43	***	***	**
C20: 0	1.55 ^d^	1.67 ^d^	3.52 ^b^	2.65 ^c^	5.02 ^a^	2.93 ^c^	0.36	***	***	***
C20: 1	5.02 ^b^	5.13 ^b^	5.14 ^b^	7.98 ^b^	14.71 ^a^	6.68 ^b^	1.07	**	**	*
C20: 3 n6	14.46 ^d^	22.64 ^bc^	15.26 ^cd^	24.02 ^b^	33.22 ^a^	17.80 ^bcd^	1.99	**	**	n.s.
C20: 4 n6	9.13 ^ab^	6.88 ^b^	9.28 ^ab^	12.12 ^a^	12.26 ^a^	5.81 ^b^	0.78	n.s.	*	**
C20: 5 n3	3.66 ^ab^	5.07 ^a^	2.97 ^ab^	4.78 ^a^	5.28 ^a^	1.63 ^b^	0.41	n.s.	**	n.s.
C22: 6 n3	13.66	14.48	14.28	17.03	16.78	16.29	0.65	n.s.	n.s.	n.s.
SFA	361.04 ^d^	447.98 ^bc^	504.62 ^ab^	355.64 ^d^	408.37 ^cd^	563.23 ^a^	23.14	n.s.	***	n.s.
MUFA	334.39 ^c^	343.46 ^c^	711.72 ^a^	265.03 ^c^	543.41 ^b^	713.36 ^a^	55.34	n.s.	***	**
PUFA	83.73 ^d^	141.93 ^ab^	120.84 ^bc^	116.24 ^bc^	153.03 ^a^	107.27 ^cd^	7.02	*	***	*
PUFA/SFA	0.23 ^b^	0.32 ^a^	0.24 ^b^	0.33 ^a^	0.37 ^a^	0.19 ^b^	0.02	n.s.	**	*
∑n − 3	21.73 ^b^	24.75 ^ab^	23.44 ^ab^	26.73 ^ab^	32.70 ^a^	31.54 ^a^	1.45	*	n.s.	n.s.
∑n − 6	62.00 ^d^	117.19 ^a^	97.41 ^b^	89.51 ^bc^	120.34 ^a^	75.73 ^cd^	6.47	n.s.	***	**
∑n − 6/∑n − 3	2.87 ^cd^	4.77 ^a^	4.16 ^ab^	3.35 ^bcd^	3.76 ^abc^	2.42 ^d^	0.26	*	*	*
Total	779.16 ^d^	933.36 ^c^	1337.18 ^a^	736.90 ^d^	1104.81 ^b^	1383.85 ^a^	76.52	*	***	*

SFA, saturated fatty acids; MUFA, monounsaturated fatty acids; PUFA, polyunsaturated fatty acids. ∑n − 3 = sum of C18: 3n−3, C20: 5n3 and C22: 6n3; ∑n − 6 = sum of C18: 2n−6, C20: 3n−6 and C20:4n6. ^abcd^ Values in the same row without a common letter are significantly different (*p* < 0.05). Sig.: significance; n.s.: not significant; * *p* < 0.05; ** *p* < 0.01; *** *p* < 0.001.

**Table 4 animals-12-01161-t004:** The commonly identified volatile compounds (ng/100 g) in LT muscle from Angus and Xiangxi yellow cattle.

Volatile Compounds	LRI	I.M	Angus			Xiangxi Yellow Cattle			s.e.m.	Sig.		
			6 M	18 M	30 M	6 M	18 M	30 M		B	M	B × M
Octanol	1073	MS + LRI	1562.68 ^c^	1766.95 ^c^	1786.42 ^c^	3221.31 ^b^	4753.63 ^a^	4513.87 ^a^	404.14	***	*	n.s.
1-Octen-3-ol	980	MS + LRI	N.D.	563.55	711.82	1344.24	2042.90	2450.20	263.20	—	—	—
Dodecanol	1476	MS + LRI	812.98	914.08	1028.85	795.94	879.51	817.37	29.04	n.s.	n.s.	n.s.
Hexadecanol	1880	MS + LRI	1084.61	1176.22	1081.27	1052.83	951.78	1059.17	42.22	n.s.	n.s.	n.s.
2-Hexyl-1-decanol	1786	MS + LRI	2523.30	2716.63	2967.30	3012.78	2806.27	2972.66	170.64	n.s.	n.s.	n.s.
2-Octyl-1-decanol	1677	MS	3314.86 ^c^	3361.78 ^c^	3203.98 ^c^	4040.57 ^b^	4166.29 ^b^	5637.14 ^a^	258.56	***	*	**
Alcohols	—	—	9298.41 ^d^	10,499.19 ^d^	10,779.63 ^d^	13,467.67 ^c^	15,600.37 ^b^	17,450.40 ^a^	895.47	***	**	n.s.
Octanal	1003	MS + LRI	2156.62	2795.93	3107.31	2941.68	3198.27	2996.77	134.93	n.s.	n.s.	n.s.
Nonanal	1105	MS + LRI	12,437.41	11,939.67	13,864.11	12,062.94	11,796.93	11,435.95	308.25	n.s.	n.s.	n.s.
Decanal	1207	MS + LRI	3936.42 ^b^	3883.30 ^b^	5383.10 ^a^	2001.75 ^c^	2050.26 ^c^	2198.34 ^c^	393.47	***	n.s.	n.s.
Undecanal	1308	MS + LRI	1827.03	1800.41	2048.18	1650.41	1915.99	1697.08	49.63	n.s.	n.s.	n.s.
Dodecanal	1409	MS + LRI	1828.89 ^c^	1869.27 ^c^	2108.36 ^b^	2141.69 ^b^	2625.82 ^a^	2651.77 ^a^	100.07	***	**	*
Tridecenal	1511	MS + LRI	643.17 ^b^	671.93 ^b^	743.96 ^b^	1561.11 ^a^	1487.35 ^a^	1435.51 ^a^	124.41	***	n.s.	n.s.
Tetradecanal	1612	MS + LRI	2080.45 ^d^	2065.69 ^d^	2541.90 ^c^	4969.85 ^b^	5527.77 ^a^	5495.24 ^a^	474.36	***	*	n.s.
Pentadecanal	1714	MS + LRI	7904.48 ^b^	9965.40 ^b^	9098.07 ^b^	15162.01 ^a^	16267.62 ^a^	15499.82 ^a^	1046.87	***	n.s.	n.s.
Hexadecanal	1816	MS + LRI	5292.71 ^e^	7460.42 ^d^	7317.41 ^d^	11584.75 ^c^	13660.54 ^b^	15454.13 ^a^	1113.10	***	***	*
Heptadecanal	1918	MS + LRI	3312.56 ^c^	3633.17 ^c^	3367.36 ^c^	6422.51 ^b^	7699.21 ^a^	7502.23 ^a^	584.24	***	*	n.s.
4,8,12-Tetradecatrienal,5,9,13-trimethyl-	1839	MS + RI	1376.66 ^c^	2742.84 ^b^	3778.69 ^a^	1528.07 ^c^	1882.98 ^c^	1514.37 ^c^	267.11	**	**	**
Octadecanal	2020	MS + LRI	952.16 ^c^	1403.26 ^b^	1520.57 ^b^	4167.96 ^a^	4336.31 ^a^	4398.71 ^a^	458.13	***	*	n.s.
Aldehydes	—	—	43,748.53 ^d^	50,231.27 ^c^	54,879.00 ^c^	66,194.69 ^b^	72,511.41 ^a^	72,217.51 ^a^	3356.49	***	**	n.s.
2-Undecanone	1295	MS + LRI	891.76 ^c^	863.95 ^c^	988.44 ^b^	816.63 ^c^	1208.38 ^a^	1313.74 ^a^	58.43	**	**	*
6,10-Ddimethylundeca-5,9-Dien-2-One	1454	MS + LRI	1922.12 ^d^	2470.47 ^c^	3477.13 ^a^	1718.02 ^d^	2891.99 ^b^	2751.12 ^bc^	181.04	n.s.	***	**
ketones	—	—	2813.87 ^c^	3334.42 ^b^	4465.57 ^a^	2534.65 ^c^	4100.37 ^a^	4064.86 ^a^	218.43	n.s.	***	*
Hexanoic acid	996	MS + LRI	N.D.	N.D.	N.D.	2856.90	3306.70	3200.74	489.59	—	—	—
Nonanoic acid	1281	MS + LRI	359.76 ^c^	316.16 ^c^	1129.82 ^b^	2400.57 ^a^	1975.30 ^a^	2589.58 ^a^	283.29	***	*	n.s.
Tetradecanoic acid	1767	MS + LRI	2597.96	2532.56	3003.73	2787.84	2239.76	1958.00	126.26	n.s.	n.s.	n.s.
Hexadecanoic acid	1967	MS + LRI	15,178.13 ^b^	13,546.74 ^b^	25,978.42 ^a^	16,443.53 ^b^	14,910.24 ^b^	13,761.61 ^b^	1351.42	*	*	**
Organic acids	—	—	18,135.84 ^cd^	16,395.46 ^d^	30,111.97 ^a^	24,488.84 ^b^	22,432.00 ^bc^	21,509.92 ^bcd^	1410.95	n.s.	*	**
Sulfurous acid, 2-ethylhexyl isohexyl ester	1488	MS	1133.18 ^c^	1606.11 ^b^	1154.14 ^c^	2029.43 ^a^	2063.72 ^a^	2228.23 ^a^	134.32	***	n.s.	*
Dibutyl phthalate phthalate	1869	MS + LRI	730.74 ^b^	858.64 ^a^	921.87 ^a^	585.54 ^c^	546.07 ^c^	729.26 ^b^	41.52	***	**	n.s.
Esters	—	—	1863.93 ^c^	2464.75 ^b^	2076.00 ^c^	2614.97 ^ab^	2609.78 ^ab^	2957.49 ^a^	114.39	**	n.s.	*
Dodecane	1199	MS + LRI	1462.54	1455.51	1546.10	1578.45	1878.32	1553.50	64.35	n.s.	n.s.	n.s.
Dodecane, 2-methyl-	1263	MS + LRI	2844.87	2918.08	3156.25	3056.76	3835.46	4111.04	189.53	n.s.	n.s.	n.s.
2-Bromo dodecane	1420	MS + LRI	1738.91	1666.24	1713.57	1805.80	2027.01	1899.76	42.48	n.s.	n.s.	n.s.
Tridecane	1299	MS + LRI	2820.92	2803.98	2633.76	3187.22	3063.63	3273.30	129.82	n.s.	n.s.	n.s.
Tetradecane	1399	MS + LRI	3039.50	2765.86	2616.46	2795.86	2825.50	2818.13	120.74	n.s.	n.s.	n.s.
Tetradecane, 5-methyl-	1463	MS + LRI	700.51 ^c^	921.07 ^b^	616.93 ^c^	1148.52 ^a^	1186.09 ^a^	1249.20 ^a^	74.25	***	*	**
Dodecane, 4,6-dimethyl-	1279	MS + LRI	3328.02 ^bc^	4555.31 ^ab^	3053.44 ^c^	5227.35 ^a^	4862.95 ^a^	5211.14 ^a^	286.16	**	n.s.	n.s.
Nonane,5-(2-methylpropyl)-	1243	MS + LRI	516.49 ^c^	749.95 ^bc^	618.19 ^c^	1029.52 ^ab^	1152.55 ^a^	1163.55 ^a^	81.89	**	n.s.	n.s.
Pentadecane	1498	MS + LRI	3655.60	2630.71	2738.06	4339.99	3848.17	3771.00	250.12	n.s.	n.s.	n.s.
Hexadecane	1508	MS + LRI	3638.01	2954.00	3153.00	4458.47	4167.34	3553.81	208.23	n.s.	n.s.	n.s.
Heptadecane	1707	MS + LRI	3988.82 ^d^	4467.90 ^c^	3806.79 ^d^	6592.67 ^b^	8476.24 ^a^	8313.08 ^a^	606.22	***	*	*
Octadecane	1798	MS + LRI	900.44	897.78	813.70	1183.77	1303.83	1310.13	63.77	n.s.	n.s.	n.s.
Hexadecane, 2,6,10,14-tetramethyl-	1751	MS + LRI	1219.59 ^d^	1914.40 ^c^	981.28 ^d^	1825.97 ^c^	3014.26 ^a^	2434.92 ^b^	209.84	***	***	*
2-Hexadecene, 3,7,11,15-tetramethyl-	1844	MS + LRI	1687.12 ^d^	1771.59 ^d^	1727.33 ^d^	2014.26 ^bc^	2966.85 ^ab^	3849.04 ^a^	260.56	**	n.s.	n.s.
Hydrocarbons	—	—	31,541.30 ^b^	32,472.35 ^b^	29,175.13 ^b^	40,244.58 ^a^	44,608.17 ^a^	44,511.47 ^a^	1976.78	***	n.s.	n.s.
1,3-Dioxolane, 2-heptyl-	1159	MS + LRI	3031.31	2717.41	1944.82	2379.02	2786.93	2115.00	153.26	n.s.	n.s.	n.s.
2(3 H)-Furanone, 5-hexyldihydro-	1364	MS + LRI	637.09 ^c^	745.65 ^c^	752.07 ^c^	1384.04 ^b^	1701.92 ^a^	1714.51 ^a^	138.97	***	**	n.s.
Heterocyclic compounds	—	—	3668.40	3463.06	2696.89	3763.06	4488.85	3829.51	103.03	n.s.	n.s.	n.s.
2, 4-Ditert-Butyl Phenol	1517	MS + LRI	3970.92 ^c^	4646.68 ^c^	4531.10 ^c^	12,622.93 ^b^	15,771.76 ^a^	16,589.06 ^a^	1644.51	***	**	**
Dioctyl ether	1684	MS + LRI	1425.05 ^e^	1653.03 ^d^	1703.15 ^d^	2841.83 ^c^	3156.70 ^b^	3639.71 ^a^	255.93	***	***	*
Others	—	—	5395.97 ^d^	6299.71 ^d^	6234.25 ^d^	15,464.76 ^c^	18,928.46 ^b^	20,228.77 ^a^	1897.79	***	***	**

LRI, linear retention indexes; I.M, identification methods. ^abcde^ Values in the same row without a common letter are significantly different (*p* < 0.05). Sig.: significance; n.s.: not significant; * *p* < 0.05; ** *p* < 0.01; *** *p* < 0.001.

**Table 5 animals-12-01161-t005:** Sensory evaluation of LT muscle from Angus and Xiangxi yellow cattle.

Sensory Test	Angus			Xiangxi Yellow Cattle			s.e.m.	Sig.		
	6 Month	18 Month	30 Month	6 Month	18 Month	30 Month		Breed	Month	B × M
Tenderness	4.50 ^a^	4.42 ^a^	3.79 ^b^	4.38 ^a^	4.25 ^a^	3.58 ^b^	0.11	n.s.	***	n.s.
Juiciness	3.59 ^b^	3.85 ^a^	3.88 ^a^	3.41 ^b^	3.94 ^a^	3.86 ^a^	0.06	n.s.	**	n.s.
Beefy flavor	3.80 ^c^	4.06 ^b^	4.08 ^b^	4.07 ^b^	4.47 ^a^	4.57 ^a^	0.08	***	**	n.s.
Abnormal flavor	2.59 ^c^	2.94 ^b^	2.83 ^bc^	3.64 ^a^	3.44 ^a^	3.42 ^a^	0.12	***	n.s.	*
Flavor liking	3.73	3.88	4.12	3.74	3.92	3.93	0.05	n.s.	n.s.	n.s.
Overall liking	4.21	4.25	4.21	3.99	4.23	4.10	0.04	n.s.	n.s.	n.s.

^abc^ Values in the same row without a common letter are significantly different (*p* < 0.05). Sig.: significance; n.s.: not significant; * *p* < 0.05; ** *p* < 0.01; *** *p* < 0.001.

## Data Availability

Data sharing is not applicable.

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
