# Peer review of "Muscle Fatty Acids, Meat Flavor Compounds and Sensory Characteristics of Xiangxi Yellow Cattle in Comparison to Aberdeen Angus"

_animals, 2022, doi:10.3390/ani12091161_

Round 1

Reviewer 1 Report

Review Animals-1671730
Muscle fatty acids, meat flavor compounds and sensory characteristics of Xiangxi yellow cattle in comparison to Aberdeen Angus by Chen et al.
The manuscript submitted for review is an important position both in terms of the quality of beef from Xiangxi yellow cattle but also in terms of the factors that shape this quality as culinary beef i.e. free amino acids, 5'-nucleotides, fatty ccids and volatile compounds. In this respect, it is certainly a noteworthy position, in the light of the current state of knowledge.
The document is publishable.
The document is well written and easy to read.
I found no methodical or logical errors.
However, it requires improvement of several elements which I have listed below.
Please specify the source of the taste activity threshold values used in your manuscript.
Line 55. Did the authors mean "Xiangxi area"?
Chapter 2.1. An important addition to the manuscript would be to provide the feed ration composition and its value taking into account the age of the animals (or breed if there were differences).
Line 110. Period after parenthesis [].
I think there were long spaces in the text (especially lines: 80, 408)
Line 164. 2 cm3 (superscript 3)
Line 166. 1 - 8 point scale (space after 1)
Chapter 3.1. It may be useful to give the most important values of the obtained results shown in Figure 1. In other words, it will be difficult to cite.
Line 204. health [15]. (period after parenthesis)
Lines 238, 266, 301, 395. Space before Sig.:
Line 239. 3.3. 5'-. nucleotides (period and space)
Line 241. [24,25]. Period after parenthesis [].
Line 260 and in the Table 2. (μg/ g) (without spaces)
Line 262. threshold. Equivalent (space after a period)
Line 304. (Legako et al. 2015) - the citation should be adjusted according to the requirements of the journal
Line 399. PC1 explained 48.55%
The "references" chapter need to be be adjusted according to the requirements of the journal. Depending on the type of work: https://www.mdpi.com/journal/animals/instructions#references

Author Response

Thank you for your correction. I have made relevant modifications and supplements. Please refer to the attached document for details .Best wishes to you.

Reviewer 2 Report

The manuscript compares the meat quality characteristics, in terms of fatty acids, flavour compounds and sensory profile, of Xiangxi yellow cattle and Aberdeen Angus at different slaughter ages. While the science was sound in terms of the methods used, the manuscript needs to be edited to improve readability, highlight novelty, and add depth. Before being considered for publication, several significant issues need to be addressed in this paper.

  • The introduction is confusing and represents different ideas not clearly connected. No hypothesis is provided and the introduction does a poor job of establishing the gap in knowledge that exists and how this study will fill it. What additional information will be studying this specific breed provide over what is already available? Authors can spend a few lines to explain clearly the reasoning behind their choice and also the gap in knowledge that this study is attempting to fill.
  • Several parts of the results and discussion sections are not accurate and need to be further improved. In particular, the discussion did not provide a complete explanation and context of the results. Throughout the discussion and the conclusion sections of the paper, the authors mention the Xiangxi yellow cattle but fail to discuss in depth the results obtained to show the uniqueness of this data in this particular breed. This paragraph must be profoundly modified.
  • In the conclusion section, the authors highlight only the results obtained. Conclusions should be rewritten to give more relevance to your results. What is the general message from your study?
  • There are numerous English and grammar mistakes throughout the manuscript that need to be corrected to improve readability. The reviewer has pointed out a few suggestions in the section below, but these are by no means all of the corrections that need to be made.

Specific comments:

line 15; 48-49: delete “cattle are important meat source”.

line 50: cattle?? Which? The Xiangxi yellow cattle?

line 51: delete “eliminated”.

line 53-55: this statement needs a reference.

line 55-56: please delete this sentence.

line 122; 149: change “as in literature” into “ according to the procedure of …”.

line 178-180: this statement is not in line with the results of figure 1. However, in my opinion, the figure should be replaced with a table, to make the results more readable. Also, the figure is of poor quality (it should be at least 300 dpi).

line 247- 248: change “Statistical analysis” into “ Results from this study revealed that…”.

line 248: change “moths” into “age at slaughter”.

line 259: 18 months

line 280: 30 months

line 381-383: please revise to improve readability.

line 401- 403;  405- 408: confusing sentences that need to be rewritten.

Figures 1 and 2. The resolution of the figures should be improved.

Author Response

(The authors gave the same response as above.)

Round 2

Reviewer 2 Report

The manuscript is well improved after revision, in particular, authors have improved the previous version both in the introduction section as well as in the results and discussion. In my opinion, the current version of the manuscript is worthy of publication on Animals.